# Reducing Defocused-Information Crosstalk to Multi-View Holography by Using Multichannel Encryption of Random Phase Distribution

Chih-Hao Chuang [1] , Chien-Yu Chen [2],*, Hsuan-Ting Chang [3], Hoang-Yan Lin [1] and Chuan-Feng Kuo [2]

1    Graduate Institute of Photonics and Optoelectronics, National Taiwan University, Taipei 106, Taiwan; zx610438@gmail.com (C.-H.C.); hoangyanlin@ntu.edu.tw (H.-Y.L.)
2    Graduate Institute of Color and Illumination Technology, National Taiwan University of Science and Technology, Taipei 106, Taiwan; n8w6mpk80@gmail.com
3    Department of Electrical Engineering, National Yunlin University of Science and Technology, Yunlin 640, Taiwan; htchang@yuntech.edu.tw
*    Correspondence: chencyue@mail.ntust.edu.tw; Tel.: +886-2-2737-6742

**Abstract:** A new optical encryption algorithm, called the random-phase encryption algorithm, is proposed in this study. When this algorithm was applied in constructing computer-generated holograms, the out-of-focus image crosstalk was effectively reduced, and the image quality was greatly enhanced. In this study, the researchers encrypted each multi-depth or multi-view random phase sub-image with the phase-locked key to generate multi-channel encryption phase information. During the reconstruction, the switch of the phase-locked key of the sub-image was found to achieve different image reconstruction effects with different views or depths. This algorithm proved to substantially reduce the out-of-focus image crosstalk and to enhance the reconstruction quality of the original computer holography without concerning the mutual interference among the information of each view for multi-view and multi-depth holograms.

**Keywords:** optical encryption algorithm; reducing defocused-information crosstalk; computer-generated holography

## 1. Introduction

At the present stage, the development of stereoscopic display technology is restricted because of the accommodation-vergence confliction, to which experts have proposed solutions such as the realization of multi-depth images, light field display technology, and computer holography with multilayer waveguide. However, different from multilayer waveguide display and light field display technology, holography reconstructs the optical information image in the space by the plane wave traveling through a spatial light modulator. For multi-depth image projection, it is still affected by information crosstalk, resulting in worsened stereoscopic perception of image and image blur [1] and limiting of the viewing angle [2]. Moreover, in terms of the presentation and watching of multi-view images (including views and depths), defocus information on the non-reconstruction plane would reduce watching quality, which could be regarded as multi-view holography crosstalk [3]. Recent research teams have proposed many corresponding solutions. For instance, Yan Li's team changed the single plane Gerchberg–Saxton (GS) algorithm into dynamic compensation GSA and adjusted weighting factors in amplitude-constrained function in each iterative operation to effectively enhance the different-depth image quality in the multiplane 2D holographic image display [4]. Different from the optimization of pure 2D images, F. Ömer Ilday's team proposed to stack 3D images with multi-layer images. However, the multi-plane information crosstalk worsened in this regard. Thus, they proposed to first process the wavefront preshaping of each piece of information with the Fresnel zone plate and orthogonality of the random vector. Then, they used orthogonality of the random

vector to reduce the crosstalk across the images to stack multi-plane information and create three-dimensional information with low information interference [5]. Although this method could enhance the computing speed, more 2D images were required to realize high-resolution 3D images or dynamic information, which required a considerable increase in the size of the computational information.

In this study, optical encryption technology is applied to solve the problem of multi-depth and multi-view image crosstalk. Optical encryption, with its uniqueness, is broadly highlighted; e.g., high speed, high parallel, and key application flexibility [6–10]. After Javidi's team first proposed double-random-phase encryption [6] in 1995, a series of optical image encryption methods were proposed, such as the use of Fractional Fourier transform [7], Fresnel transform [8], discrete wavelet transform [9], and phase contrast [10]. The maturity of optical encryption system development is apparent.

As a result, based on the previously proposed 3D modified Gerchberg-Saxton algorithm [11–13], optical encryption is included in this study with random phase as the key to reduce out-of-focus image interference and enhance image watching quality. The phase-locked key-encrypted multi-depth or multi-view image information is capable of realizing the display of multi-view holographic contents by switching decryption keys. The proposed phase encryption technology with angle multiplexing enables multi-view watching and effectively removes out-of-focus image crosstalk. The objective evaluation of the proposed algorithm reveals the significant enhancement of overall image quality.

## 2. Methods

### 2.1. Algorithm Multiplexing

The Modified Gerchberg–Saxton algorithm (MGSA) used in this study could transform phase-only function (POF) with optical wavelength or imaging position [12,13]. The POFs with distinct encryption conditions on the Fresnel transform (FrT) plane were combined as a single POM (phase-only mask). The study applied position multiplexing to realize the multi-depth image CGH display. Using the x-axis as the axis of rotation as an example, the geometric coordinate of Fresnel diffraction between the corresponding imaging tilted plane and input plane [13] is shown in Figure 1. After the image phase information traveled through the Fresnel diffraction, the output image parallel to the reference image would be shown on the output plane. The original field signal placed on the input plane, after Fresnel diffraction, could acquire the diffraction field image $G(x,y)$ on the output plane, which is parallel to the reference plane. To obtain the distance between a point on the tilted output plane and the corresponding point on the input plane, a calculation, shown as Equation (1), was required.

$$r_x = \sqrt{(x_0 - x')^2 + (y_0 - y'\cos\theta_x)^2 + (z - y'\sin\theta_x)^2} \tag{1}$$

where $\theta x$ stands for the angle of $x$ axial rotation. When the output plane revolves the $x$ axis and is not parallel to the input plane, only the Fresnel diffraction of the object on the tilted plane [14,15] would be calculated in order to parallel the diffracted output plane with the reference plane. To acquire the tilted Fresnel equations, the signal $g(x_0, y_0)$ on the input plane is transformed, shown as Equation (2) below:

$$G_x(x', y') = \exp\left(i\frac{2\pi}{\lambda}r_x\right) \iint\limits_{-\infty}^{\infty} g(x_0, y_0) \exp\left[i\frac{\pi}{\lambda r_x}\left(x_0^2 + y_0^2\right)\right]$$
$$\times \exp\left\{-i\frac{2\pi}{\lambda r_x}[x_0 x' + y_0 y'\cos\theta_x + (z_0 - r_x)y'\sin\theta_x]\right\}dx_0 dy_0 \tag{2}$$

where $\lambda$ is the wavelength and $z_0$ is the distance between the center point on the tilted input plane and the center point on the output plane. Assuming $N$ pieces of the target image being $g_n(x', y')$, $n = 1{\sim}N$, each image corresponds to the different angles of the inclination $\theta x$. The individual phase information $\psi_{\theta_{xn}}(x_0, y_0)$ was acquired through MGSA calculation.

When the value above is substituted in Equation (1), it could be simplified as Equation (3), shown below:

$$G_x(x', y') = \text{FrT}\{\exp[j\psi_{\theta_{xn}}(x_0, y_0)]; \lambda; z_0; \theta_{xn}\} \\ = \hat{g}_{\theta_{xn}}(x', y')\exp[j\varphi_{\hat{g}_{xn}}(x', y')] \tag{3}$$

where $\varphi_{\hat{g}_n}(x', y')$ is the individual phase signal of $\psi'_{\theta_{xn}}(x_0, y_0)$, $n = 1 \sim N$ after FrT, shown in Figure 2. Individual phases were integrated into a single piece of phase-only mask (POM), as shown in Equation (4):

$$\exp[j\psi_T^{\theta_x}(x_0, y_0)] = \exp\left\{\sum_{n=1}^{N}\exp[j\psi'_{\theta_{xn}}(x_0, y_0)]\right\} \tag{4}$$

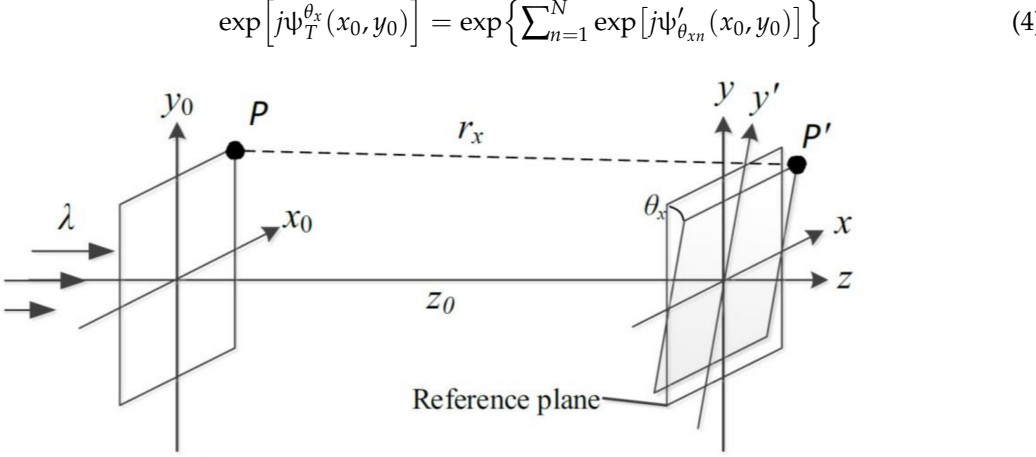

**Figure 1.** FrT space coordinates relationship diagram of the input plane revolving the x axis.

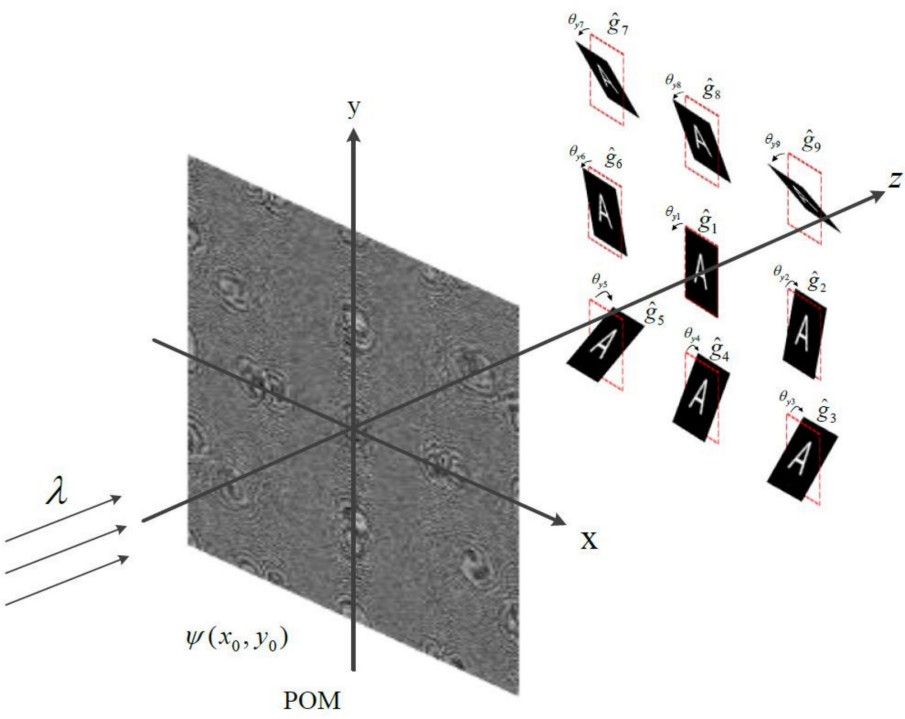

**Figure 2.** Multi-channel image optical architecture diagram of the single phase revolving the x axis.

### 2.2. Phase-Locked Algorithm

The multi-depth and multi-view holographic display could be realized with algorithm multiplexing. However, a holographic display is different from watching natural objects with different distances, because the front images would not cover the rear images. In the holographic display, coaxial optical information with different depths would cause

mutual interference, as in Figure 3a. In a multi-view holographic display, crosstalk is apparent when the angular interval of images is small, as in Figure 3b. To effectively resolve the issue above, the phase encryption was introduced under the framework of the 3D MGSA. The individual multi-depth or multi-view sub-images were encrypted with different phase-locked keys to generate phase-locked images. When decrypting, the corresponding phase-locked keys were used to present the complete sub-images. Such a method essentially reduced the crosstalk caused by the out-of-focus image in multi-view and multi-depth image reconstruction and enhanced the image watching quality. The phase-locked algorithm could be illustrated in two parts: the encryption and the decryption. Figure 4 shows the entire phase encryption–decryption process, where the left part shows the encryption and the right part displays the decryption. The complete process can be seen in Figures 5 and 6. Figure 5 shows the encryption process. Figure 6 shows the decryption process.

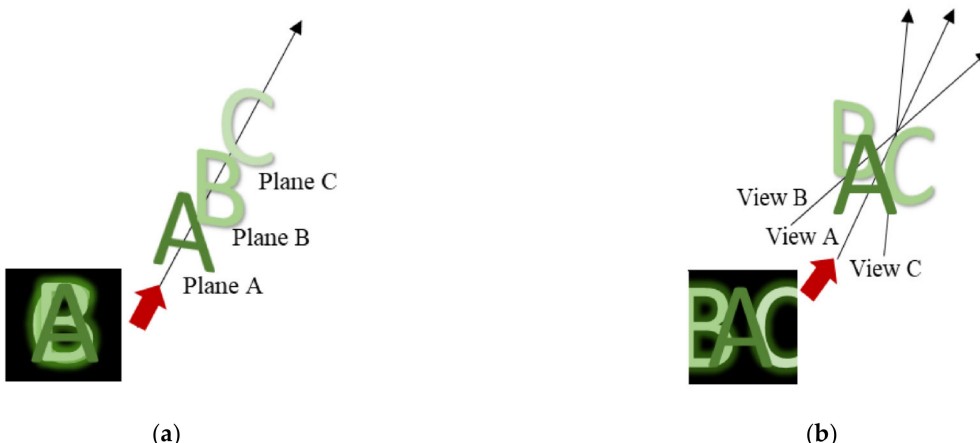

(**a**)            (**b**)

**Figure 3.** Crosstalk in holographic display in (**a**) multi-depth and (**b**) multi-view.

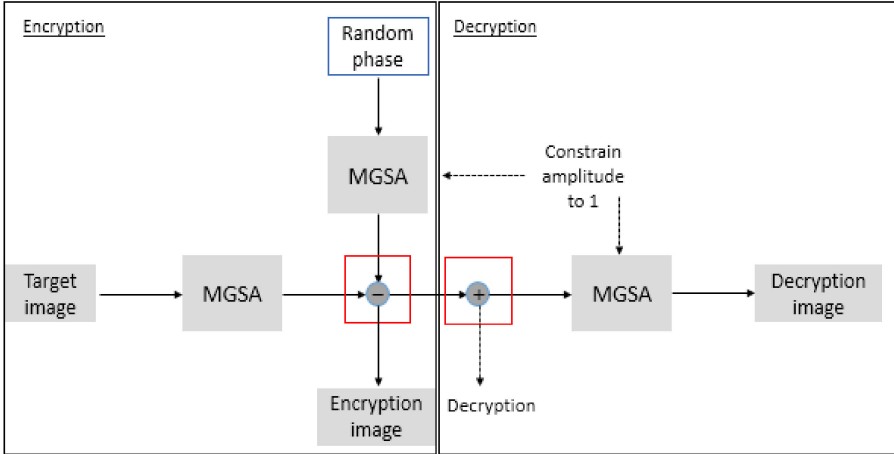

**Figure 4.** Phase-locked algorithm flow chart.

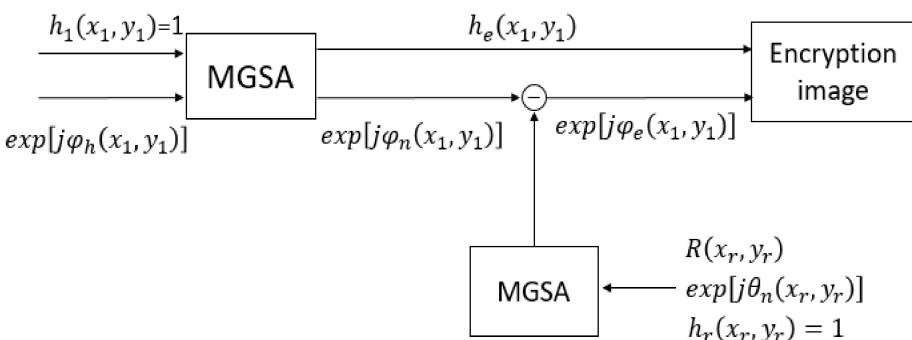

**Figure 5.** Encryption flow chart.

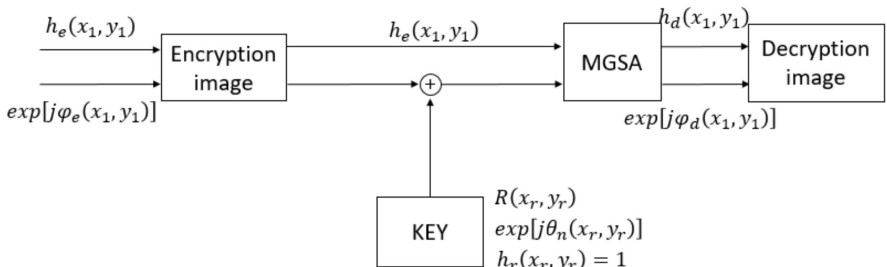

**Figure 6.** Decryption flow chart.

Encryption process

- Step 1: Generate a random phase signal $\varphi_h(x_1, y_1)$ and set the initial amplitude = 1. Multiply both to acquire the image amplitude $h_e(x_1, y_1)$ and the phase signal $\varphi_n(x_1, y_1)$.
- Step 2: Calculate phase-locked key $\theta_n(x_r, y_r)$ with MGSA to modulate the amplitude to 1 and only retain the phase information.
- Step 3: Subtract phase signal $\varphi_n(x_1, y_1)$ from random phase signal $\theta_n(x_r, y_r)$ to get the encryption image $\varphi_e(x_1, y_1)$.

Equations (5) and (6) are Step 1, Equations (7) and (8) are Step 2, and Equation (9) is Step 3. The equations for the encryption process are shown below:

$$G(x_0, y_0) = FrT\{1 \times exp[j\varphi_h(x_1, y_1)]\} \tag{5}$$

$$g(x_1, y_1) = IFrT\{G(x_0, y_0)\} \tag{6}$$

$$R(x_1, y_1) = FrT\{1 \times exp[j\theta_n(x_r, y_r)]\} \tag{7}$$

$$r(x_r, y_r) = IFrT\{G(x_1, y_1)\} \tag{8}$$

$$En(x_1, y_1) = angle\{g(x_1, y_1)\} - angle\{r(x_r, y_r)\} \tag{9}$$

Decryption process

- Step 1: Add the encryption image phase signal $\varphi_e(x_1, y_1)$ and the random phase signal $\theta_n(x_r, y_r)$.
- Step 2: Multiply the encryption image amplitude $h_d(x_1, y_1)$ and the decryption phase signal $\varphi_e(x_1, y_1)$ for the MGSA calculation and modulate the amplitude to 1.
- Step 3: Obtain the decryption image $\varphi_e(x_1, y_1)$.

Equation (10) is Step 1, where the phase-locked key is not changed, Equation (11) is Step 2, and Equation (12) is Step 3. Equations for the decryption process are shown below:

$$Reg(x_1, y_1) = En(x_1, y_1) - angle\{r(x_r, y_r)\} \tag{10}$$

$$A(x_0, y_0) = FrT\{Reg(x_1, y_1)\} \tag{11}$$

$$\text{De}(x_1, y_1) = IFrT\{A(x_0, y_0)\} \tag{12}$$

### 2.3. Assessment Approaches of Image Reconstruction Quality

In this study, qualities of diffraction imaging results and simulated imaging results were assessed, and the criteria included relative diffraction efficiency (RDE), root mean square error (RMSE), and signal to noise ratio (SNR) [16–18]. Before the aforementioned assessments, signal reconstruction and area reconstruction needed to be defined, as in Figure 7. Signal reconstruction was the signal area of the reconstructed target image, while area reconstruction was the entire area for reconstruction. The reconstruction image area was assumed as X × Y, including signal area and noise of the reconstructed target image, and the intensity of reconstruction signal and noise was assumed to be $I_s$ and $I_N$, respectively.

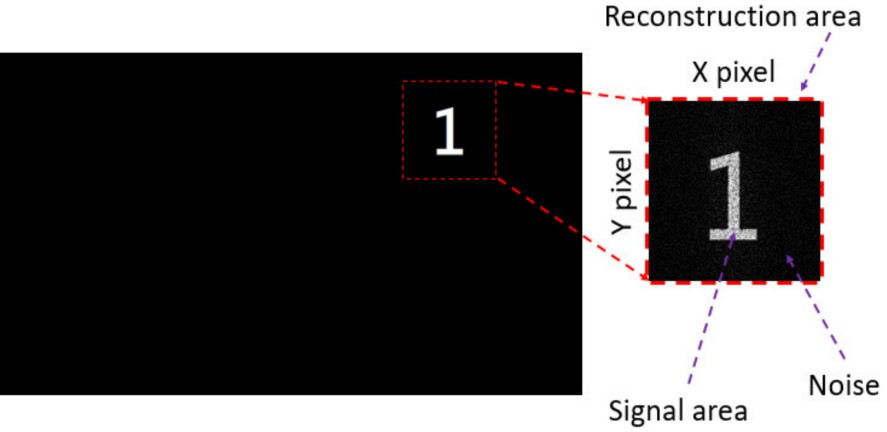

**Figure 7.** Schematic diagram of area reconstruction, signal reconstruction, and noise.

Relative diffraction efficiency was used to determine whether or not the background noise affected image quality, where the intensity sum $\sum I_s$ of each pixel in the signal reconstruction area was the numerator, and the intensity sum $\sum(I_s + I_N)$ of each pixel in the reconstruction area was the denominator. If the result was somewhere near 100%, there was nearly no background noise and the reconstruction image quality was good, with the equation as demonstrated in Equation (13) [16]:

$$I_{RDE} = \frac{\sum I_s}{\sum(I_s + I_N)} \times 100\% \tag{13}$$

Root mean square error (RMSE) is an objective assessment, often used for defining reconstruction image qualities. It was used to calculate the mean difference between each pixel in diffraction reconstruction images and original images, mainly to judge the similarity of the signal distribution between diffraction reconstruction images and original images. If the RMSE value was closer to 0, it revealed the higher similarity of diffraction reconstruction images and original images and lower distortion, as in Equation (14) [17]:

$$RMSE = \sqrt{\sum \frac{I_N^2}{XY}} \tag{14}$$

Signal to noise ratio is the ratio of diffraction reconstruction signal intensity to noise intensity, which was used to compare the intensity level between the target signal and the background signal. Decibel (dB) normally serves as the unit; if the SNR was higher than

0 dB, it represented more reconstruction signals than noises, which could be an indicator of the diffraction reconstruction image being clearly reconstructed, as in Equation (15) [18]:

$$SNR(dB) = 10 \times log_{10} \frac{I_s}{I_N} \tag{15}$$

## 3. Experiments and Results

In this study, the input image resolution was 110 × 110, and the resolution of output phase information was 1920 × 1080. The reconstruction was calculated in two parts. In the first part, the multi-depth image, the digits "1", "2", and "3" were placed at the reconstruction distance 0.2 m, 0.3 m, and 0.4 m for phase-locked encryption transform. By switching the corresponding phase-locked key, the image with corresponding depth could be effectively reconstructed. Figure 8 shows the reconstruction simulation results. Figure 8a,b display the differences. With a phase-locked algorithm, coaxial reconstruction with distinct depth essentially reduced the interference caused by coaxial out-of-focus image information, as in Figure 8a. The image assessment results are shown in Table 1.

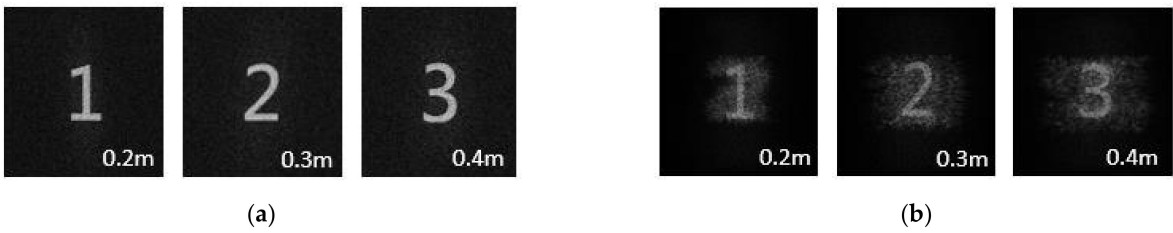

(a) (b)

**Figure 8.** Results of coaxial reconstruction with distinct depth (**a**) with phase-locked algorithm and (**b**) without phase-locked algorithm.

**Table 1.** Assessment results of reconstruction images simulated with multi-depth phase encryption algorithm.

|  | Picture | RDE | RMSE | SNR |
|---|---|---|---|---|
| MGSA algorithm | 1 | 69.22% | 0.0262 | 3.52 dB |
|  | 2 | 76.62% | 0.0246 | 5.16 dB |
|  | 3 | 62.17% | 0.0286 | 3.72 dB |
| Phase encryption algorithm | 1 | 82.56% | 0.0180 | 6.75 dB |
|  | 2 | 92.99% | 0.0122 | 11.23 dB |
|  | 3 | 91.66% | 0.0132 | 10.41 dB |

In the second part, the multi-angle image, the digits "1", "2", and "3" were placed at the reconstruction distance 0.2 m, with the encryption angles of 42°, 45°, and 48°, respectively, for the phase-locked encryption transform. By switching the corresponding phase-locked keys in the reconstruction, the images with corresponding angles could be effectively reconstructed. Figure 9 shows the reconstruction simulation results. Noticeable differences can be seen in Figure 9a,b; with the phase-locked algorithm, the interference of coaxial out-of-focus image information was drastically reduced in the reconstruction with the same depth but different angles, as shown in Figure 9a. The image assessment results are shown in Table 2.

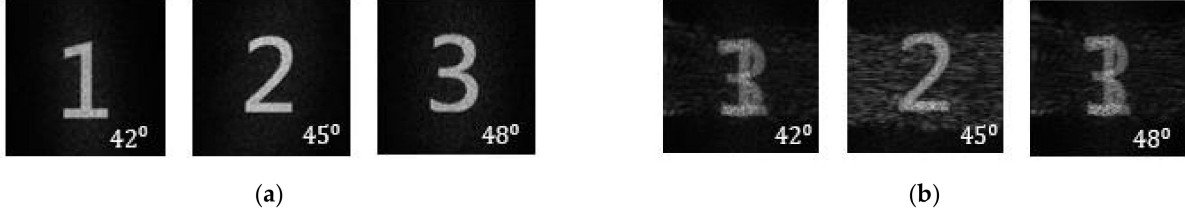

(**a**)                                                        (**b**)

**Figure 9.** Reconstruction with same depth but different view (**a**) with phase-locked algorithm and (**b**) without phase-locked algorithm.

**Table 2.** Assessment results of reconstruction images simulated with multi-angle phase encryption algorithm.

|  | View | RDE | RMSE | SNR |
|---|---|---|---|---|
| MGSA algorithm | 42° | 39.71% | 0.0315 | 2.157 dB |
|  | 45° | 41.72% | 0.0689 | 1.4513 dB |
|  | 48° | 40.23% | 0.0284 | 1.1362 dB |
| Phase encryption algorithm | 42° | 94.0% | 0.013 | 11.957 dB |
|  | 45° | 91.4% | 0.0258 | 10.266 dB |
|  | 48° | 92.46% | 0.023 | 10.884 dB |

The optical experiment was also conducted to verify the phase-locked algorithm's effectiveness further. The optical setup is depicted in Figure 10. It comprises a collimated light source, a beam splitter (BS), two spatial light modulators (SLMs), and a charge coupled device (CCD). The collimated light source has a wavelength of 532 nm. These two SLMs are pure-phase-modulated from Jasper Display Corp (JD8554), with 1920 × 1080 pixels, 256 grey levels, and a pixel size of 6.4 μm × 6.4 μm. The phase-locked key and encryption image phase are added to the two SLMs accordingly. When the collimated light propagates through the BS and SLM1, the light can be modulated by the phase-locked key loaded on SLM1. Then, the modulated light can illuminate SLM2 after the reflection of SLM1. The images with the corresponding phase-locked keys could be effectively reconstructed by switching the corresponding phase-locked keys in the reconstruction. The results of the experiments are slightly different from the simulations due to factors such as the system alignment and the tolerance of image capturing, but the results of the phase-locked algorithm optimization can be identified. Noticeable differences can be seen in Figures 11a,b and 12a,b; with the phase-locked algorithm, the interference of coaxial out-of-focus image information was reduced in the reconstruction, as shown in Figures 11a and 12a.

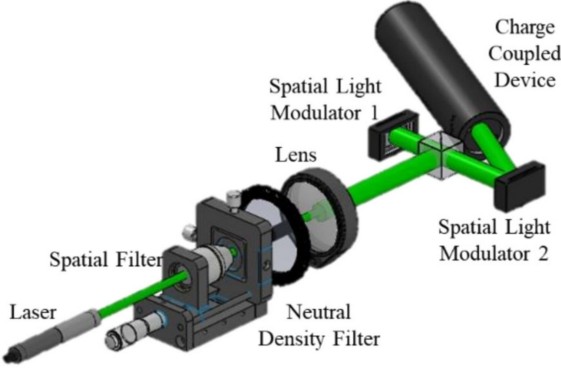

**Figure 10.** Schematic of optical setup.

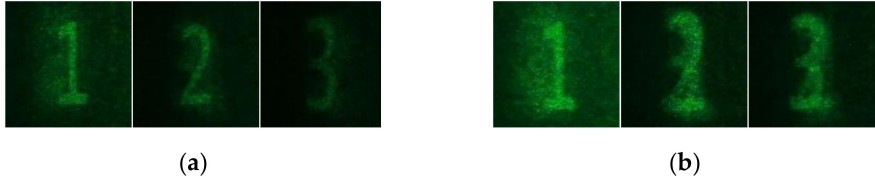

Figure 11. Experimental results of coaxial reconstruction with distinct depth (**a**) with phase-locked algorithm and (**b**) without phase-locked algorithm.

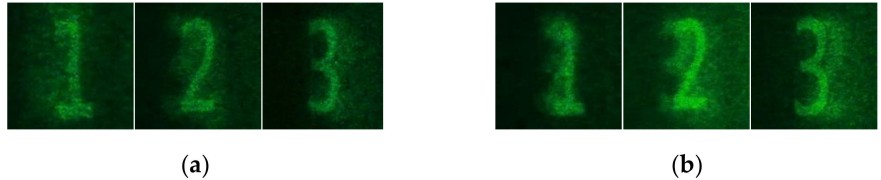

Figure 12. Experimental results of reconstruction with same depth but different view (**a**) with phase-locked algorithm and (**b**) without phase-locked algorithm.

## 4. Discussion

The results from Table 1 regarding the multi-depth encryption performance reveal that the phase encryption algorithm proposed in this study by encrypting and decrypting images through random keys could effectively enhance image qualities and reduce the interference of out-of-focus images. In the same optical axis, the multi-depth phase encryption result, the RDE results reveal that the image reconstruction quality is greatly enhanced, to at least 80% of the quality level, due to the reduction of out-of-focus image interference. When examining the distortion of diffraction reconstruction images based on the RMSE value, the reduced RMSE value indicates lower image distortion. Moreover, if the SNR values are compared, the noise of different reconstruction depths can be found to decrease, i.e., reducing image interference. To verify the results in Figure 8, the method proposed in this study, compared to the previous MGSA algorithm, could effectively reduce out-of-focus image crosstalk.

In terms of the performance of angle/depth encryption, Table 2 shows that the phase encryption algorithm could enhance the quality of multi-view images. From the RDE results, the reduction of out-of-focus image interference tremendously enhances the image reconstruction quality to more than 80% of the quality level. Judging from the distortion of the diffraction reconstruction image with RMSE values, the results reveal that the reduced RMSE value indicates lower image distortion. Finally, the comparison of SNR values demonstrates that the noise of each angle was obviously reduced, i.e., reducing image interference. Figure 9b reveals that when calculating multi-angle images with the MGSA algorithm, images with different views interfere in focusing images. On the contrary, the method proposed in this study could effectively reduce out-of-focus image crosstalk.

Based on the analysis of the more detailed intensity distribution, out-of-focus images have great effects on the results. As shown in Figure 13a, the impacts of the out-of-focus image appear to be serious. Figure 13b–d display the reconstruction results with phase-locked encryption, which is shown to effectively remove the effects of out-of-focus images. Figure 14a shows the result of the side view, where the effect of the out-of-focus imaging actually exists. Figure 14b–d displays the result of side view reconstruction with phase-locked encryption, where the effects of the out-of-focus image are significantly decreased.

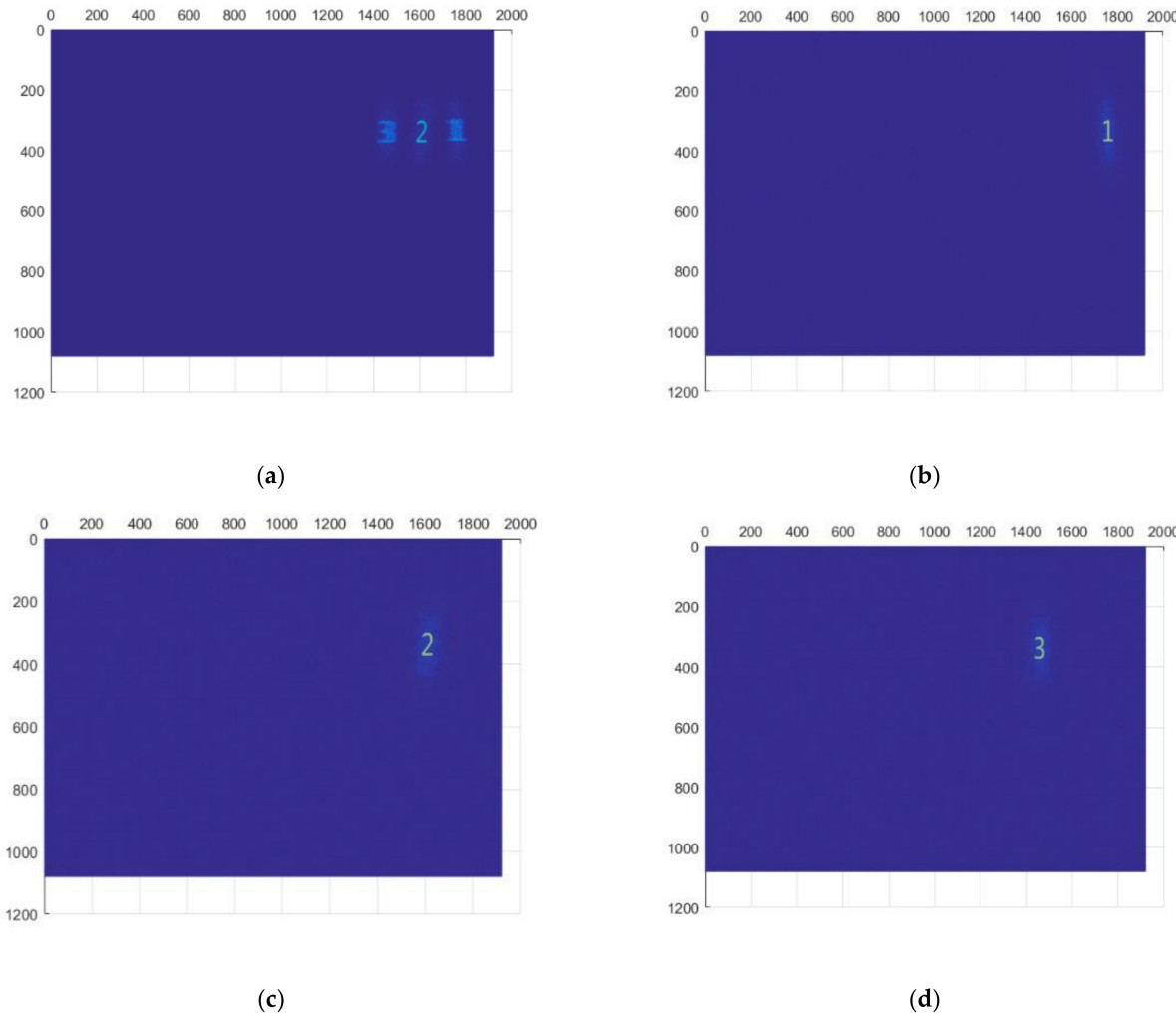

**Figure 13.** Front view analysis results (**a**) without phase-locked algorithm and (**b**–**d**) with phase-locked algorithm.

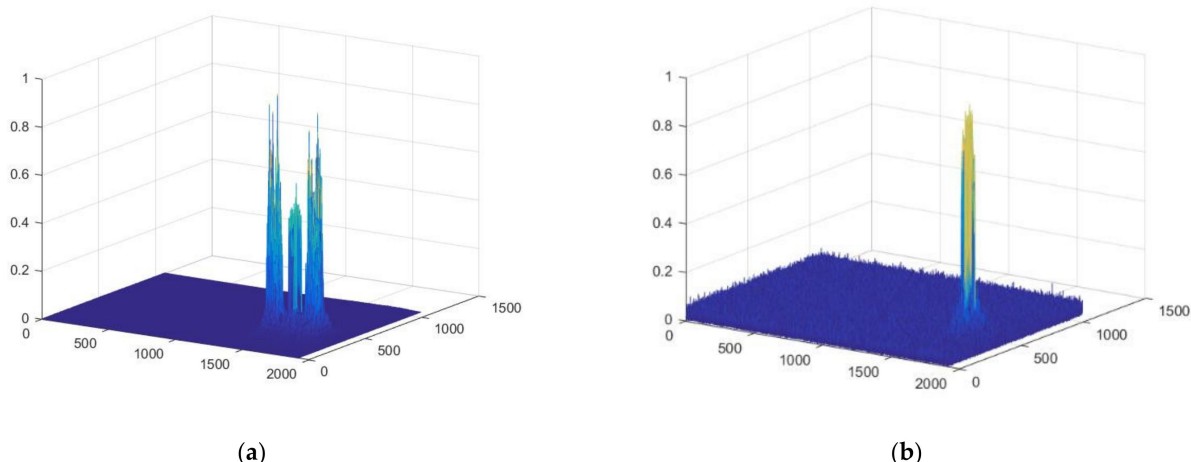

**Figure 14.** *Cont.*

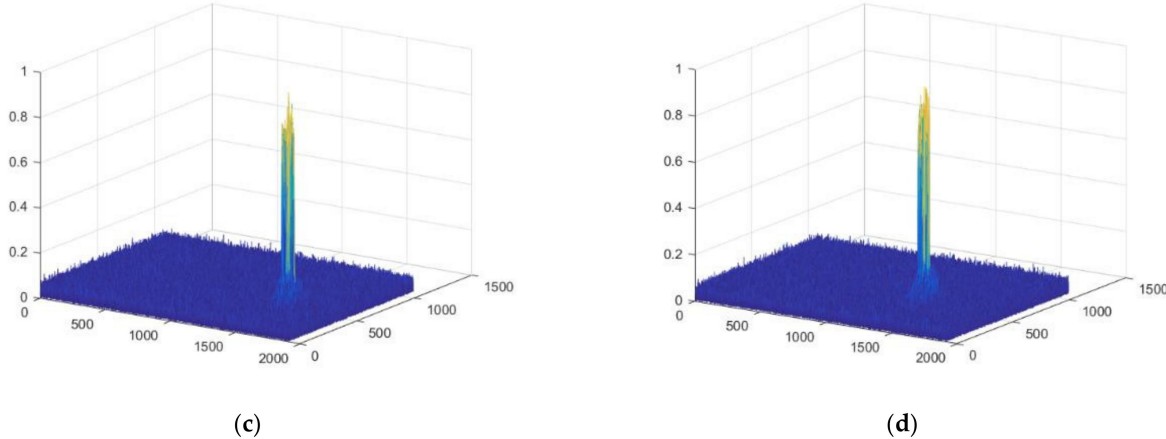

(**c**)                    (**d**)

**Figure 14.** Side view analysis results (**a**) without phase-locked algorithm and (**b**–**d**) with phase-locked algorithm.

## 5. Conclusions

A phase encryption algorithm was proposed in this study. It encrypted each multi-depth or multi-view sub-image with the corresponding phase-locked keys and used the corresponding phase-locked keys for encryption to decrypt and to show the entire image. This could be applied to the encryption of multi-view and multi-depth images. According to reconstruction images simulated with the algorithm, the reduced impact of the out-of-focus images essentially enhanced the image quality. Overall, this study verifies the principle and the practicability of the aforementioned method; the proposed phase encryption algorithm was also capable of reducing out-of-focus image crosstalk and was demonstrated in the image quality assessment results.

**Author Contributions:** Data curation, C.-F.K.; Formal analysis, C.-H.C. and C.-F.K.; Investigation, C.-Y.C.; Methodology, C.-H.C. and H.-T.C.; Project administration, C.-Y.C. and H.-Y.L.; Software, C.-H.C. and H.-T.C.; Supervision, C.-Y.C. and H.-Y.L. All authors have read and agreed to the published version of the manuscript.

**Funding:** This research was funded by the Ministry of Science and Technology of Taiwan under contract No. 110-2218-E-011-009-MBK&110-2221-E-011-149.

**Institutional Review Board Statement:** Not applicable.

**Informed Consent Statement:** Not applicable.

**Data Availability Statement:** The data presented in this study are available on request from the corresponding author.

**Conflicts of Interest:** The authors declare no conflict of interest.

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
