# Peer review of "Reducing Defocused-Information Crosstalk to Multi-View Holography by Using Multichannel Encryption of Random Phase Distribution"

_applsci, doi:10.3390/app12031413_

Round 1

Reviewer 1 Report

The authors have reported a method by which the de-focused image can be effectively removed in the holographic reconstruction. Random phase combined with phase locking has been used as the technique to realize the purpose. The simulation results demonstrate the effectiveness of the proposed idea. My main concern is the following,

The purpose of removing the de-focused images in a 3D reconscruction is not clear. In fact the presence of the defocused images in other depth planes actually provides a better 3D preception and a viewing experience cloase to a real 3D scene. So, I believe preserving the defocused images is good for a 3D display. On the other hand, isolating the images from other planes will be good for digital holographic microscopic reconstructions.

Other comments
1. The usage of English is poor at many pleases. I advice to make an English correction.

2. The conclusion not being supported by any experimental results makes a weak case.

3. Many of the images are appear blurred and it is better to increase the resolution of the images.

Author Response

Reviewer #1, The authors have reported a method by which the de-focused image can be effectively removed in the holographic reconstruction. Random phase combined with phase locking has been used as the technique to realize the purpose. The simulation results demonstrate the effectiveness of the proposed idea. My main concern is the following,

Concern # 1: The purpose of removing the de-focused images in a 3D reconscruction is not clear. In fact the presence of the defocused images in other depth planes actually provides a better 3D preception and a viewing experience close to a real 3D scene. So, I believe preserving the defocused images is good for a 3D display. On the other hand, isolating the images from other planes will be good for digital holographic microscopic reconstructions.

Author response:

Thank you for your comment. In the reconstruction of the hologram, the reconstruction quality of the image will be affected by information that does not fulfill the decryption conditions. This algorithm was proved to substantially reduce the out-of-focus image crosstalk and to enhance the reconstruction quality of the original computer holography without concerning the mutual interference among the information of each view for multi-view and multi-depth holograms.

Concern # 2: The usage of English is poor at many pleases. I advise to make an English correction.

Author response:

Thank you for your comment. We have revised the manuscript accordingly.

Concern # 3: Thank you for your reminder. The conclusion not being supported by any experimental results makes a weak case.

Author response:

Thank you for your reminder. This study focuses on the method specificity of the algorithm, so only the simulated results are presented. However, we tried to implement the experiment according to the reviewer's request. The results of the experiments are slightly different from the simulations due to some factors such as system setup and image capturing, but the results of the phase-locked algorithm optimization can be identified. Please see attachment Figures 1&2. We revised the conclusion in the manuscript.

Concern # 4: Many of the images are appear blurred and it is better to increase the resolution of the images.

Author response:

Thank you for your reminder. The figures in the article have been updated and confirmed.

Reviewer 2 Report

In the manuscript, the authors proposed a new optical encryption algorithm to reduce out-of-focus images crosstalk and improve images quality when reconstructing computer-generated holograms.

I read the manuscript, but I found it very difficult to understand. In my opinion, the manuscript fails in clarity, therefore I suggest a revision of the language and of the style of the presentation. For example, I found lines from 150 to 157 and lines from 80 to 84 completely incomprehensible. Similar, the lines from 25 to 28.

In conclusion, I suggest considering the manuscript for publication after a complete rewriting.

Author Response

Reviewer#2, In the manuscript, the authors proposed a new optical encryption algorithm to reduce out-of-focus images crosstalk and improve images quality when reconstructing computer-generated holograms.

Concern # 1: I read the manuscript, but I found it very difficult to understand. In my opinion, the manuscript fails in clarity, therefore I suggest a revision of the language and of the style of the presentation. For example, I found lines from 150 to 157 and lines from 80 to 84 completely incomprehensible. Similar, the lines from 25 to 28.

Author response:

Thank you for your comment. We have revised the content in the manuscript.

Lines150 to 157

“In this study, qualities of diffraction imaging results and simulated imaging results were assessed and the criteria included relative diffraction efficiency (RDE), root mean square error (RMSE), and signal to noise ratio (SNR) [16-18]. Before the aforementioned assessments, signal reconstruction and area reconstruction needed to be defined first. Signal reconstruction was the signal area of the reconstructed target image while area reconstruction was the entire area for reconstruction. The reconstruction image area was assumed X × Y, including signal area and noise of the reconstructed target image, and the intensity of reconstruction signal and noise was assumed  ad , respectively.”

Lines 80 to 84

“where ?? stands for the angle of x axial rotation. When the output plane revolved x axis and was not parallel to the input plane, only the Fresnel diffraction of the object on the tilted plane [14,15] would be calculated in order to parallel the diffracted output plane with the reference plane. To acquire the tilted Fresnel equations, the signal  on the input plane was transformed, shown as Equation (2) below:”

Lines 25 to 28.

“At the present stage, the development of stereoscopic display technology is restricted because of the accommodation-vergence confliction, to which experts proposed solutions, such as the realization of multi-depth images, light field display technology, and computer holography with multilayer waveguide.”

Concern # 2: In conclusion, I suggest considering the manuscript for publication after a complete rewriting.

Author response:

Thank you for your reminder. We have revised the manuscript accordingly.

Round 2

Reviewer 1 Report

I find the authors have addressed most of the concerns. Since the effect of phase-locked algorithm is clearly visible in the experimental results, I strongly recommend the authors to include those results and its optical setup. The paper may be accepted after this inclusion.

Author Response

We appreciate your efforts and comments on our study. We have revised the manuscript according to your comments.

Reviewer 2 Report

In the present work, the authors proposed a new optical encryption algorithm to reduce out-of-focus images crosstalk and improve images quality when reconstructing computer-generated holograms. Here, the authors compared the performance of the proposed algorithm with the performance provided by an MGSA algorithm by demonstrating the better performances of the first one.

In my opinion, the authors well-addressed the referees' comments reported after the first round, and the manuscript, in the present form, appears better organized and easier to understand. In the end, the conclusions are well-supported by the theoretical results, therefore, despite the minor issues attached, I retain the present work suitable for publication in the Applied Sciences journal.

Author Response

We appreciate your efforts and comments on our study.
